# Learning Compact Semantic Information for Incomplete Multi-View Missing Multi-Label Classification

Jie Wen [1]  Yadong Liu [1]  Zhanyan Tang [1]  Yuting He [2]  Yulong Chen [1]  Mu Li [1]  Chengliang Liu [1]

## Abstract

Multi-view data involves various data forms, such as multi-feature, multi-sequence and multimodal data, providing rich semantic information for downstream tasks. The inherent challenge of incomplete multi-view missing multi-label learning lies in how to effectively utilize limited supervision and insufficient data to learn discriminative representation. Starting from the sufficiency of multi-view shared information for downstream tasks, we argue that the existing contrastive learning paradigms on missing multi-view data show limited consistency representation learning ability, leading to the bottleneck in extracting multi-view shared information. In response, we propose to minimize task-independent redundant information by pursuing the maximization of cross-view mutual information. Additionally, to alleviate the hindrance caused by missing labels, we develop a dual-branch soft pseudo-label cross-imputation strategy to improve classification performance. Extensive experiments on multiple benchmarks validate our advantages and demonstrate strong compatibility with both missing and complete data.

## 1. Introduction

Multimodal learning aims to integrate data from different modalities (such as images, text, speech, etc.) to enhance the model's understanding and decision-making capabilities. This approach has shown significant potential in fields such as medical diagnosis (Qiu et al., 2022; Luo et al., 2023; Lin et al., 2025), wearable devices (Wen et al., 2022; Fang et al., 2024), sentiment analysis (Soleymani et al., 2017; Zhang

et al., 2023), and multimedia information retrieval (Fang et al., 2022). Not only does multimodal data with significant differences exhibit rich representational power, but multi-feature data from the same sample also holds substantial representational capabilities. The learning paradigm for performing machine learning or data analysis tasks based on such multimodal or multi-feature data is unified within the theoretical framework of multi-view learning (Zhang et al., 2019). And the success of this framework in downstream tasks is largely attributed to extracting and leveraging multi-view consistent semantic information (Hwang et al., 2021; Zeng et al., 2023; Fang et al., 2023).

In traditional single-label classification tasks, it is typically assumed that each sample corresponds to a unique label. However, in real-world applications, almost all samples may be associated with multiple labels. For example, in autonomous driving tasks, images captured by cameras may contain multiple entities and be associated with several labels. Multi-Label Learning (MLL) tasks have attracted increasing attention due to their strong practical significance (Zhou et al., 2012), which aims to predict all possible relevant labels for an unlabeled example. Recent extensive research has witnessed significant progress of MLL in various practical applications, e.g., document classification (Fallah et al., 2023), emotion recognition (Zhang et al., 2021; Ameer et al., 2022), and image annotation (Chen et al., 2019).

Combining the characteristics of the two tasks mentioned above, multi-view multi-label classification (MVMLC) methods (Wu et al., 2019; Zhang et al., 2020) aims to leverage multi-view semantic information of samples as much as possible to enhance the performance of multi-label classification. However, in real-world data collection processes, failures often occur inevitably due to various reasons. For example, in medical diagnosis, missing diagnostic reports often occur due to privacy concerns or economic factors. Additionally, manual annotation of labels may lead to incomplete tagging. This leads to the necessity of incomplete Multi-view Missing Multi-Label Classification (iM3C), which focuses on scenarios where some views and labels are missing.

Considering the incomplete multi-view problem, Zhao et al. proposed a method named CDMM (Zhao et al., 2021),

[1]School of Computer Science and Technology, Harbin Institute of Technology, Shenzhen, 518000 China [2]Department of Biomedical Engineering, Case Western Reserve University, Cleveland, OH 44106 USA. Correspondence to: Chengliang Liu <liucl1996@163.com>.

*Proceedings of the $42^{nd}$ International Conference on Machine Learning*, Vancouver, Canada. PMLR 267, 2025. Copyright 2025 by the author(s).

which explores multi-view consistency and diversity while employing an ensemble learning strategy to obtain the final results. To alleviate insufficient available label information, Zhao et al. proposed a method named LVSL based on the structural consistency between the view and the label space (Zhao et al., 2022). Li et al. approached the problem from the perspective of incomplete multi-label learning, leveraging the prediction matrix to capture global and local label structures to address the issue of incomplete labels (Ma & Chen, 2021). The above three methods have made significant contributions to the fields of multi-view learning or multi-label classification. However, they are unable to fully address the iM3C problem. Therefore, Tan et al. proposed a method named iMVWL, which can simultaneously learn a shared subspace from incomplete views with missing labels (Tan et al., 2018). Li et al. proposed a method named NAIM3L, which constructs a composite index matrix involving available views and known label information to mitigate negative impacts (Li & Chen, 2021). In recent years, deep learning has achieved significant development (Yang et al., 2023b). In addition to above traditional methods, Liu et al. introduce deep incomplete instance-level contrastive learning in the iM3C task to extract multi-view features (Liu et al., 2023). To further leverage supervised information, Xie et al. applied dual graph constraints to view-level and label-level embeddings, to generate missing labels through uncertainty-aware strategies (Xie et al., 2024b). Considering the coupling between multi-view consistency and multi-label classification tasks, a method named SIP is proposed to construct label prototypes in a data-driven manner, which further extracts task-relevant information via the information bottleneck approach (Liu et al., 2024).

The success of these methods is closely tied to the extraction and learning of multi-view consistency information (i.e., shared information). To take it a step further, we assume that almost all the information relevant to downstream classification tasks is shared among views (Federici et al., 2020; Tsai et al., 2020). Despite the significant progress made by existing contrastive learning based methods in extracting consistent semantic features, we observe that the effect of contrastive learning in aggregating multi-view latent features remains limited on incomplete data, showing the inadequacy of contrastive learning in dealing with incomplete multi-view data. Therefore, in this paper, we propose a mutual information enhancement strategy based on a Mixture of Experts (MoE) framework, which compresses the shared information in cross-view joint representations by maximizing cross-view mutual information between shared representations and the raw data. In addition, the insufficient of weak supervisory information in missing multi-label data also hinders the further improvement of multi-label classification performance. In response to this, we adopt a dual-branch soft pseudo-label cross-imputation strategy to alleviate the negative impact of missing supervisory information.

Overall, for iM3C task, we propose a novel consistent semantic representation learning framework, named as **COME**. Compared to existing methods, our contributions can be summarized as follows:

- We propose a multi-view semantic consistency enhancement strategy to learn compact multi-view shared information, which effectively alleviates the performance degradation caused by incomplete contrastive learning.

- Compared to existing works, we adopt a dual-branch co-training approach, dynamically generating soft pseudo-labels for missing multi-label imputation.

- Our **COME** shows good adaptability to arbitrary view and label incompleteness. Extensive experimental results, both in complete and incomplete cases, demonstrate the superiority and robustness of our method.

## 2. Preliminary and Motivation

### 2.1. Problem Definition

For the iM3C task, we define variable set $\{\mathbf{x}^{(v)}\}_{v=1}^{\mathbf{m}}$ as any sample with $\mathbf{m}$ observations, where $x_i^{(v)} \in \mathbb{R}^{d_v}$ is $i$-th sampling of variable $\mathbf{x}^{(v)}$ and $d_v$ denotes the dimensionality of $v$-th view. And we define $\mathbf{y} \in \{0,1\}^{\mathbf{c}}$ as the incomplete label set, where $\mathbf{c}$ is the number of categories. Taking into account the impact of missing views and missing labels, we define $\mathcal{V}$, with $|\mathcal{V}| \leq \mathbf{m}$, as the set of available views. Specifically, $\{\mathbf{x}^{(v)}\}_{v \in \mathcal{V}}$ ($\mathbf{X}^{\mathcal{V}}$ for short) denotes the views that can be observed. Similarly, we also define $\mathcal{G}$, with $|\mathcal{G}| \leq \mathbf{c}$, to denote the set of known categories. We define $n$ as the number of samples. Our goal is to learn the multi-view joint representation $\mathbf{z}$ from $\mathbf{X}^{\mathcal{V}}$ which contains all the semantic consistency information that can be used to accurately predict the corresponding categories.

### 2.2. Sufficiency of Multi-View Shared Representation

Suppose for each sample we just have two views, and $\mathbf{z}^{(1)}, \mathbf{z}^{(2)}$ are latent representations of $\mathbf{x}^{(1)}, \mathbf{x}^{(2)}$, and we introduce a variable $T$ to represent the classification-related information. Since we suppose all the task-relevant information contained by multi-view shared representation, inspired by (Federici et al., 2020; Tsai et al., 2020), we introduce the following redundancy assumption:

**Assumption 2.1.** $\mathbf{x}^{(1)}$ and $\mathbf{x}^{(2)}$ share the same semantic information, which is sufficient for prediction: $I(\mathbf{x}^{(1)}; T) = I(\mathbf{x}^{(2)}; T)$. That is to say, if $\mathbf{z}^{(1)}$ satisfy $I(\mathbf{x}^{(1)}; \mathbf{x}^{(2)}|\mathbf{z}^{(1)}) = 0$, then $\mathbf{z}^{(1)}$ contains all the task-relevant information for prediction: $I(\{\mathbf{x}^{(1)}, \mathbf{x}^{(2)}\}; T) = I(\mathbf{x}^{(1)}; T) = I(\mathbf{z}^{(1)}; T)$.

In multi-view applications, the representations of all views are combined to fully leverage the shared semantic information. Furthermore, we introduce the semantic sufficiency proposition for the multi-view joint representation $\mathbf{z}$:

**Proposition 2.2.** *If $\mathbf{z}$ contains all the shared information among views, $\mathbf{z}$ is sufficient for prediction.*

From Proposition 2.2, we have $I(\mathbf{z}; T) = I(\mathbf{x}^{(1)}; T) = I(\mathbf{x}^{(2)}; T)$. It should be noted that if $\mathbf{z}$ contains all the information from $\mathbf{x}^{(1)}$ and $\mathbf{x}^{(2)}$, it inherently includes all the information required for downstream tasks. However, in practical applications, the redundant information contained in $\mathbf{z}$ will lead to performance degradation. Thus, we should compress the non-shared information within the joint representation as much as possible, i.e., pursuing the minimal sufficient shared representation cross all view:

**Definition 2.3.** (Minimal and Sufficient Shared Representation) The joint representation $\mathbf{z}$ is minimal and sufficient for $\mathbf{x}^{(1)}$ and $\mathbf{x}^{(2)}$ if and only if $I(\mathbf{x}^{(1)}, \mathbf{x}^{(2)}) = I(\mathbf{z}, \mathbf{x}^{(1)}) = I(\mathbf{z}, \mathbf{x}^{(2)})$.

### 2.3. Insufficient Multi-view Contrastive Learning

Empirical evidence shows that self-supervised learning (SSL) methods, even without utilizing any downstream supervisory information during training, can still learn embeddings that generalize well to a wide range of downstream tasks. For example, SimCLR (Chen et al., 2020) defines a contrastive loss and uses augmented images as self-supervised signals, allowing the trained model to perform effectively on various tasks. In previous frameworks, self-supervised signals were often generated through data augmentation techniques, such as word masking (Devlin, 2018) or image rotation (Gidaris et al., 2018). In multi-view contrastive learning, we consider that variables belonging to the same sample but from different views should share semantic consistency information, while variables from different samples may not (Tian et al., 2020; Pan & Kang, 2021; Liu et al., 2023).

Existing studies have demonstrated the positive correlation between contrastive learning and the maximization of interview mutual information (Tsai et al., 2020; Wang et al., 2022). That is to say, the objective of contrastive learning is equal to $\max I(\mathbf{z}^{(1)}; \mathbf{z}^{(2)})$. From multi-view data $\{\mathbf{x}^{(1)}, \mathbf{x}^{(2)}\}$ and their latent representations $\{\mathbf{z}^{(1)}, \mathbf{z}^{(2)}\}$, based on the Data Processing Inequality (Thomas & Joy, 2006) and Markov chain, we have:

$$I(\mathbf{x}^{(1)}, \mathbf{x}^{(2)}) \geq I(\mathbf{z}^{(1)}, \mathbf{x}^{(2)}) \geq I(\mathbf{z}^{(1)}, \mathbf{z}^{(2)}) \quad (1)$$

The gap between $I(\mathbf{x}^{(1)}, \mathbf{x}^{(2)})$ and $I(\mathbf{z}^{(1)}, \mathbf{z}^{(2)})$ will be small enough when the model capacity is sufficient and the training data is sufficient. Ideally, contrastive learning can compress non-shared information and maximize

the proportion of shared information in the representation. However, we observed that shared representations learned through contrastive learning on incomplete data are insufficient. As shown in Fig. 1, we perform experiments on two popular multi-view multi-label datasets Corel5k and Pascal07 to illustrate the performance degradation of inadequate contrastive learning. The ratio of missing views is set to 50% and the missing instances do not participate in the construction of positive-negative pairs in contrastive learning loss. We can observe that, compared to the case of complete views, the contrastive learning running on incomplete data leads to a worse classification performance. As we know, the core idea of contrastive learning is pull together the positive sample pairs and push apart the negative samples (Khosla et al., 2020). In the case of missing views, the contrastive learning is less effective in pushing negative pairs away and pulling positive pairs closer. To address the inadequacy of contrastive learning under missing multiview data, we propose a MoE-based mutual information enhancement framework, which aims to learn the minimal sufficient representation of multi-view data by compressing redundant information and maximizing cross-view mutual information.

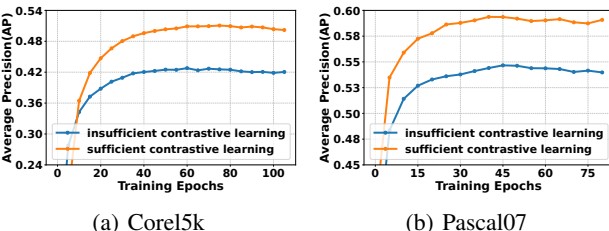

(a) Corel5k          (b) Pascal07

*Figure 1.* Impact of contrastive loss on Average Precision (AP) in Corel5k and Pascal07 datasets under incomplete and complete multi-view data.

### 2.4. Pseudo Label Learning

In the iM3C task, labels are also incomplete and previous methods leverage incomplete label indicator to mask the missing part in the training process. To reduce the negative impact of missing labels, inspired by some SSL methods (Lee et al., 2013; Berthelot et al., 2019; Sohn et al., 2020; Xu et al., 2021), an intuitive strategy is to assign pseudo-labels to samples with missing labels. Pseudo-labeling is originally proposed for the semi-supervised training of deep neural networks (Lee et al., 2013). Some methods (Berthelot et al., 2019; Sohn et al., 2020) combine consistency regularization or entropy regularization to improve the quality of pseudo-labels. A common SSL method named FixMatch (Sohn et al., 2020) aims to select the most probable label as the pseudo-label, known as Top-1 strategy. However, even without considering correctness, it is obvious that this method will neglect multiple true labels. In Fig. 2 (a), for the Pascal07 dataset, the pseudo-labels generated using the

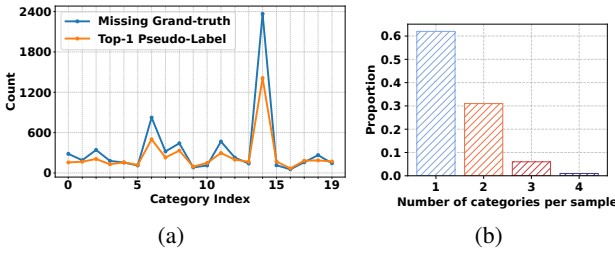

(a)                                   (b)

*Figure 2.* (a) Top-1 pseudo-labels and missing ground-truth labels at Pascal07 dataset. (b) Label count distribution across samples in Pascal07 dataset.

above Top-1 generation strategy are less than the actual number of missing labels. An alternative choice is to select the Top-k probable labels as pseudo-labels, but in practice it is difficult to set k. As shown in Fig. 2 (b), the number of true positive labels of samples in Pascal07 dataset is not a constant number. The label count distribution across samples for other datasets can be found in Appendix E. Based on this, the simple label generation strategy will inevitably introduce false positive labels or neglect true positive labels.

Furthermore, considering the class-imbalance gap between different classes (Yang et al., 2021; Fang et al., 2025), some studies propose to generate pseudo-label through complex methods, including the class-aware method (Xie et al., 2024a), the class-threshold based method (Xiao et al., 2024), and the label prototype-based method (Yang et al., 2023a). These methods directly use the generated hard pseudo-labels to train the model, treating the generated pseudo-labels as ground-truth. Since pseudo-labels are not entirely accurate, the use of hard pseudo-labels may cause the model to deviate, further amplifying errors and leading to a vicious cycle. To address this problem, we propose a dual-branch soft pseudo-label cross-imputation strategy to reduce the hindering effect of multi-label missing on the multi-label classification.

## 3. Method

In this section, we introduce our method in three parts: incomplete multi-view contrastive learning; compact shared information representation learning based on mutual information maximum; dual-branch soft pseudo-label generation and multi-label classification.

### 3.1. Incomplete Multi-view Contrastive Learning

According to the analysis in Section 2, we first establish an optimization objective to maximize mutual information between views, thereby promoting multi-view consistent semantic representation learning. Given any two views, the optimization objective can be expressed as:

$$\max \sum_{v \in \mathcal{V}} \sum_{t \neq v} I(\mathbf{z}^{(v)}; \mathbf{z}^{(t)}) \tag{2}$$

For achieving the maximization in Eq. (2), we adopt the incomplete multi-view contrastive learning method proposed by (Liu et al., 2023), then we have following contrastive loss:

$$\mathcal{L}_{CTR} = -\sum_{v \in \mathcal{V}} \sum_{t \neq v} \log \frac{\exp(\mathbf{f}(z_i^{(v)}, z_i^{(t)}))}{\exp(\mathbf{f}(z_i^{(v)}, z_i^{(t)})) + \mathbb{S}_{neg}} \tag{3}$$

where $\mathbb{S}_{neg} = \sum_{r=v,t} \sum_{j=1, j \neq i}^{n} \exp(\mathbf{f}(z_i^{(v)}, z_j^{(r)}))$, $z_i^{(v)}$ means the sampling from distribution of $\mathbf{z}^{(v)}$, and $\mathbf{f}$ is a function to measure the cosine similarity between sample-pairs. To do this, we adopt the view-specific stochastic encoder to approximate the distribution $p(\mathbf{z}^{(v)}|\mathbf{x}^{(v)})$. Similar to previous work (Shi et al., 2019; Wen et al., 2020), we also propose to factorise the joint variational posterior as a combination of unimodal posteriors, using a MoE to model $p(\mathbf{z}|\mathbf{X}^{\mathcal{V}})$, therefore we have:

$$p(\mathbf{z}|\mathbf{X}^{\mathcal{V}}) = \sum_{v \in \mathcal{V}} \frac{1}{|\mathcal{V}|} p(\mathbf{z}^{(v)}|\mathbf{x}^{(v)}) \tag{4}$$

### 3.2. Compact Shared Representation Learning

Reviewing the analysis in Section 2.3, we know that optimizing learning objective (2) with multi-view contrastive learning alone is insufficient for compression of shared information. Moreover, we also need to avoid the damage of over-compression for information validity, i.e., the learned joint representation $\mathbf{z}$ lacks sufficient information for prediction. Based on this, we propose to maximize the mutual information between the original data and the joint representation:

$$\max I(\mathbf{X}^{\mathcal{V}}; \mathbf{z}) \tag{5}$$

Obviously, Eq. (5) requires that the cross-view joint representation maintains mutual information maximization with the original data of all views. It ensures that the cross-view joint representation $\mathbf{z}$ further learns the shared information across all available views.

For mutual information $I(\mathbf{X}^{\mathcal{V}}; \mathbf{z})$ in Eq. (5), we can obtain the variational lower bound as follows:

$$\begin{aligned} &I(\mathbf{X}^{\mathcal{V}}; \mathbf{z}) \\ &= \int \int p(\mathbf{X}^{\mathcal{V}}, \mathbf{z}) \log \frac{p(\mathbf{X}^{\mathcal{V}}|\mathbf{z})}{p(\mathbf{X}^{\mathcal{V}})} d\mathbf{X}^{\mathcal{V}} d\mathbf{z} \\ &\geq \mathbb{E}_{\mathbf{X}^{\mathcal{V}} \sim p(\mathbf{X}^{\mathcal{V}})}\big[\int p(\mathbf{z}|\mathbf{X}^{\mathcal{V}}) \log q(\mathbf{X}^{\mathcal{V}}|\mathbf{z}) d\mathbf{z}\big] \end{aligned} \tag{6}$$

where $q(\mathbf{X}^{\mathcal{V}}|\mathbf{z})$ is an additional distribution introduced to approximate $p(\mathbf{X}^{\mathcal{V}}|\mathbf{z})$. And then, based on the multi-view conditional independence assumption, conditional distribution $q(\mathbf{X}^{\mathcal{V}}|\mathbf{z})$ can be decomposed into $q(\mathbf{X}^{\mathcal{V}}|\mathbf{z}) = \prod_{v \in \mathcal{V}} q(\mathbf{x}^{(v)}|\mathbf{z})$. Combined with Eq. (4), we can rewrite

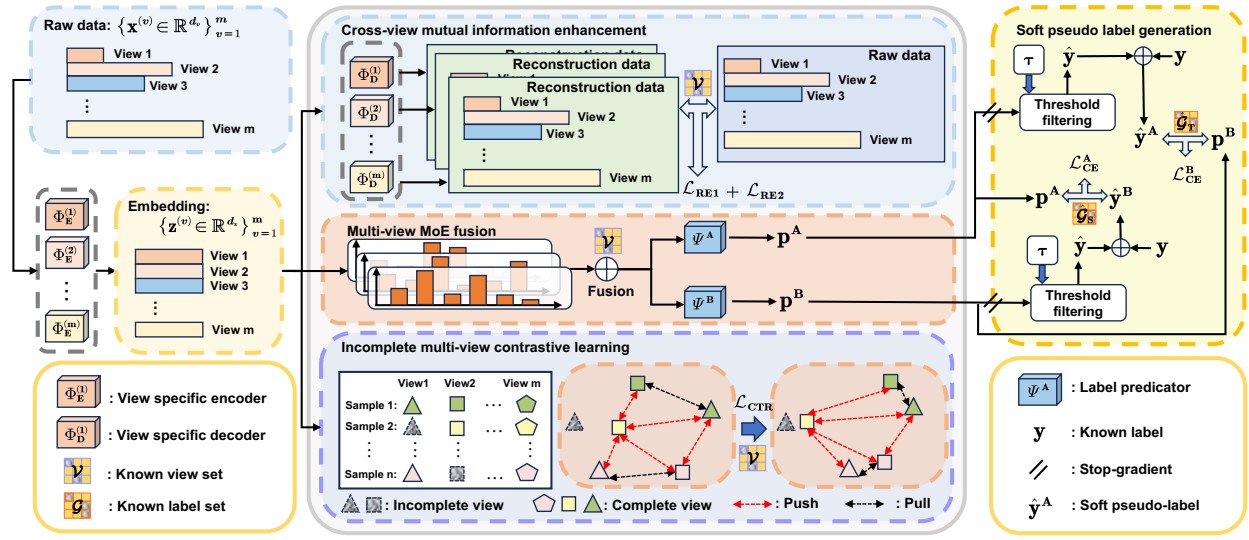

*Figure 3.* Our main framework of **COME**, consists of incomplete multi-view contrastive learning, multi-view mutual information enhancement, and dynamic soft pseudo-label generation.

the variational lower bound in Eq. (5) as follows:

$$
\mathbb{E}_{\mathbf{X}^{\mathcal{V}} \sim p(\mathbf{X}^{\mathcal{V}})} \Big[ \int p(\mathbf{z}|\mathbf{X}^{\mathcal{V}}) \log q(\mathbf{X}^{\mathcal{V}}|\mathbf{z}) d\mathbf{z} \Big]
$$

$$
= \frac{1}{|\mathcal{V}|} \sum_{v \in \mathcal{V}} \mathbb{E}_{\mathbf{x}^{(v)} \sim p(\mathbf{x}^{(v)})} \Big[ \int p(\mathbf{z}|\mathbf{x}^{(v)}) \log q(\mathbf{x}^{(v)}|\mathbf{z}) d\mathbf{z} \Big]
$$

$$
+ \frac{1}{|\mathcal{V}|} \sum_{v \in \mathcal{V}} \sum_{u \in \mathcal{V}, u \neq v} \mathbb{E}_{\mathbf{x}^{(v)} \sim p(\mathbf{x}^{(v)})} \Big[ \int p(\mathbf{z}|\mathbf{x}^{(v)}) \log q(\mathbf{x}^{(u)}|\mathbf{z}) d\mathbf{z} \Big]
$$

$$(7)$$

The full derivation of Eq. (6) and Eq. (7) is given in Appendix A. Here, we use $q(\mathbf{x}^{(v)}|\mathbf{z})$ as an approximation of $p(\mathbf{x}^{(v)}|\mathbf{z})$, which is implemented by a stochastic decoder with constant standard deviation as 1. Specifically, we set $q(\mathbf{x}^{(v)}|\mathbf{z}) = \mathcal{N}(\mathbf{x}^{(v)}|\mu^{(v)}(\mathbf{z}), \mathbf{I})$, where $\mu^{(v)}$ is the mean mapping function. Combined with the characteristics of Gaussian distribution, we have the equivalent of the optimization objective: $\max \mathbb{E}_{p(\mathbf{z}|\mathbf{x}^{(v)})}[\log q(\mathbf{x}^{(v)}|\mathbf{z})] \iff \max -\mathbb{E}_{p(\mathbf{z}|\mathbf{x}^{(v)})}[\|\mathbf{x}^{(v)} - \mu^{(v)}(\mathbf{z})\|_2^2]$. Thus, it is easy to get the following reconstruction loss for maximizing Eq. (7):

$$
\mathcal{L}_{RE} = \lambda_1 \sum_{v \in \mathcal{V}} \|\mathbf{x}^{(v)} - \mu^{(v)}(\mathbf{z}^{(v)})\|_2^2 +
$$
$$
\lambda_2 \sum_{v \in \mathcal{V}} \sum_{u \in \mathcal{V}, u \neq v} \|\mathbf{x}^{(u)} - \mu^{(u)}(\mathbf{z}^{(v)})\|_2^2 \quad (8)
$$
$$
= \lambda_1 \mathcal{L}_{RE1} + \lambda_2 \mathcal{L}_{RE2}
$$

where $\lambda_1$ and $\lambda_2$ are trade-off coefficients. Then, we combine the reconstruction loss and the contrastive loss to achieve the goal of minimal sufficient representation learning:

$$
\mathcal{L}_{MSR} = \lambda_1 \mathcal{L}_{RE1} + \lambda_2 \mathcal{L}_{RE2} + \beta \mathcal{L}_{CTR} \quad (9)
$$

where $\beta$ is the penalty coefficients for contrastive loss.

### 3.3. Dual-Branch Soft Pseudo-label Generation and Multi-Label Classification

Previous works directly use the pseudo-labels generated by the model as supervision for training in the next epoch. However, we argue that using the current network's high-confidence predictions as ground truth provides limited benefit to the network. It may also lead to overfitting on simple samples, as the model's performance bottleneck does not lie in the classification of easily distinguishable samples. Moreover, this method of using pseudo-labels amplifies the misleading effect of incorrect pseudo-labels, and errors will continue to accumulate during training. To address this, we propose a dual-branch soft pseudo-label cross-imputation strategy to efficiently generate reliable pseudo-labels, thereby further enhancing classification performance.

Specifically, we first use a dual-branch model to generate and utilize pseudo-labels, employing two independent but structurally identical models, **A** and **B**, i.e., **A**: $\{\mathbf{x}^{(v)}\}_{v=1}^m \to \mathbf{z} \to \mathbf{p}^{\mathbf{A}}$, **B**: $\{\mathbf{x}^{(v)}\}_{v=1}^m \to \mathbf{z} \to \mathbf{p}^{\mathbf{B}}$, where $\mathbf{p}^{\mathbf{A}}$ and $\mathbf{p}^{\mathbf{B}}$ are the predicted logic scores by model **A** and **B**. By doing this, we build two independent prediction models as two branches. In light of the limitation of the pseudo-label self-generation strategy in Section 2.4, we propose to leverage the external pseudo-label to improve the optimization of the model. Briefly, our core motivation is

to have two branches generate pseudo-labels for each other for the next round of iteration. To be specific, pseudo-label generation process from model **A** is as follows:

$$\hat{\mathbf{y}}^{\mathbf{A}} = \begin{cases} \mathbf{None}, & \text{if } \mathbf{c} \notin \mathcal{G} \text{ and } \mathbf{p}_{\mathbf{c}}^{\mathbf{A}} < \tau \\ \mathbf{p}_{\mathbf{c}}^{\mathbf{A}}, & \text{if } \mathbf{c} \notin \mathcal{G} \text{ and } \mathbf{p}_{\mathbf{c}}^{\mathbf{A}} \geq \tau \\ \mathbf{y}_{\mathbf{c}}, & \text{if } \mathbf{c} \in \mathcal{G} \end{cases} \quad (10)$$

where $\hat{\mathbf{y}}^{\mathbf{A}}$ denotes the pseudo-label from model **A**. In addition, we need to incorporate the generated soft pseudo-labels into the known label set and adjust the known label set $\mathcal{G}$ corresponding to $\hat{\mathbf{y}}^{\mathbf{A}}$ and $\hat{\mathbf{y}}^{\mathbf{B}}$ to get $\hat{\mathcal{G}}^{\mathbf{A}}$ and $\hat{\mathcal{G}}^{\mathbf{B}}$, respectively. To enhance the reliability soft pseudo-labels, we propose to build a dynamic threshold adjustment strategy. During the initial phases of training, a more stringent threshold is employed to ensure the high quality of pseudo-labels, thereby minimizing the risk of introducing noise. As the training progresses, the threshold is gradually relaxed to extend the range of generated pseudo-labels, striking a balance between accuracy and error:

$$\tau = \max(\tau_l, \tau_h - e \times \theta) \quad (11)$$

where $\tau_l$ denotes the lower bound of the threshold range and $\tau_h$ represents the initial largest threshold, $\theta$ is the threshold decay rate and $e$ denotes the $e$-th epoch. With the two generated soft pseudo-labels $\hat{\mathbf{y}}^{\mathbf{A}}$ and $\hat{\mathbf{y}}^{\mathbf{B}}$, we simply swap them as the supervised signal for the next training epoch of the other branch. For model **A**, the cross-entropy loss is as following:

$$\mathcal{L}_{CE} = \frac{1}{|\hat{\mathcal{G}}^{\mathbf{B}}|} \sum_{j \in \hat{\mathcal{G}}^{\mathbf{B}}} [(1 - \hat{\mathbf{y}}_j^{\mathbf{B}}) \log(1 - \mathbf{p}_j^{\mathbf{A}}) \\ + \hat{\mathbf{y}}_j^{\mathbf{B}} \log \mathbf{p}_j^{\mathbf{A}}] \quad (12)$$

Our two-branch networks generate pseudo-labels for each other, and for each one, this strategy helps it to obtain extra supervisory information from outside. The training approach and parameter update strategy for both branches are same but independent. Since the structure of the two branches is the same, we simply take the prediction of model **A** as the final result. Finally, we simply sum the objective functions of each component to obtain the following overall optimization objective:

$$\mathcal{L} = \lambda_1 \mathcal{L}_{RE1} + \lambda_2 \mathcal{L}_{RE2} + \beta \mathcal{L}_{CTR} + \mathcal{L}_{CE} \quad (13)$$

where $\lambda_1$, $\lambda_2$ and $\beta$ are hyper-parameters. For ease of understanding, we provide the training process of model **A** in Algorithm 1.

## 4. Experiments

### 4.1. Experimental Settings

**Datasets**: In line with previous works (Tan et al., 2018; Liu et al., 2023), we conduct experiments on five multi-view

---

**Algorithm 1** Training process of **COME**

1: **Input:** Incomplete multi-view data $\{\mathbf{x}^{(v)}\}_{v=1}^m$, known view set $\mathcal{V}$, incomplete multi-label $\mathbf{y} \in \{0,1\}^c$, known label set $\mathcal{G}$.
2: **Output:** Trained parameters of model **A**.
3: **Initialization:** Initialize the parameters of the model **A** and set hyper-parameters ($\lambda_1$, $\lambda_2$, $\beta$, and training epochs E)
4: **for** $e = 1, \dots, E$ **do**
5:    Compute samples latent representation $\{\mathbf{z}^{(v)}\}_{v=1}^m$ on each available view by encoders.
6:    Reconstruct available views by decoders.
7:    Calculate the incomplete multi-view contrastive loss, inter-view reconstruction loss and intra-view reconstruction loss by Eq. (9), and then obtain $L_{MSR}$.
8:    Fusion multi-view representation of samples by Eq. (4) and get multi-view joint representation $\mathbf{z}$.
9:    Obtain $\hat{\mathbf{y}}^{\mathbf{B}}$ from model **B** by Eq. (10), and obtain the corresponding new known label set $\hat{\mathcal{G}}^{\mathbf{B}}$.
10:   Calculate the classification loss $L_{CE}$ of model **A** by Eq. (12), and compute total loss of model **A**: $\mathcal{L} = L_{CE} + L_{MSR}$
11:   Update the parameters of model **A** with $\mathcal{L}$.
12: **end for**

---

multi-label datasets, i.e., Corel5k (Duygulu et al., 2002), Pascal07 (Everingham et al., 2010), ESPGame (Von Ahn & Dabbish, 2004), IAPRTC12 (Grubinger et al., 2006), and Mirflickr (Huiskes & Lew, 2008). Each dataset comprises six distinct feature types: GIST, HSV, DenseHue, DenseSIFT, RGB, and LAB. The detailed information about the five datasets and the data masking schemes that used to emulate real-word scenarios can be found in the Appendix B.

**Comparison methods**: In our experiments, we select eight popular methods for comparison with our **COME**. The selected methods include: CDMM (Zhao et al., 2021), DM2L (Ma & Chen, 2021), LVSL (Zhao et al., 2022), iMVWL (Tan et al., 2018), NAIM3L (Li & Chen, 2021), DICNet (Liu et al., 2023), UPDGD-Net (Xie et al., 2024b), SIP (Liu et al., 2024). However, not all methods are capable of handling incomplete views and labels, necessitating modifications to certain approaches. We applied specific modifications to different models, and further technical specifics can be found in Appendix C.

**Evaluation metrics**: To ensure consistency with previous work, we employed six popular performance metrics: Average Precision (AP), Hamming Loss (HL), Ranking Loss (RL), Area Under the ROC Curve (AUC), OneError (OE), and Coverage (Cov). To better present the results, we use 1-RL, 1-HL, 1-OE, 1-Cov as the final result for demonstration ensuring that higher values indicate better performance across all six metrics.

*Table 1.* Experimenntal results of nine methods on five datasets with 50% missing-view rate and 50% missing-label rate (the bottom right digit is the standard deviation).

| Data | Metric | CDMM | DM2L | LVSL | iMVWL | NAIM3L | DICNet | UPDGD-Net | SIP | **COME** |
|---|---|---|---|---|---|---|---|---|---|---|
| Corel5K | AP | $0.354_{0.004}$ | $0.262_{0.005}$ | $0.342_{0.004}$ | $0.283_{0.008}$ | $0.309_{0.004}$ | $0.381_{0.004}$ | $0.413_{0.004}$ | $0.418_{0.009}$ | $0.432_{0.009}$ |
| | 1-HL | $0.987_{0.000}$ | $0.987_{0.000}$ | $0.987_{0.000}$ | $0.978_{0.000}$ | $0.987_{0.000}$ | $0.988_{0.000}$ | $0.987_{0.000}$ | $0.988_{0.000}$ | $0.988_{0.000}$ |
| | 1-RL | $0.884_{0.003}$ | $0.843_{0.002}$ | $0.881_{0.003}$ | $0.865_{0.005}$ | $0.878_{0.002}$ | $0.882_{0.004}$ | $0.903_{0.003}$ | $0.911_{0.003}$ | $0.917_{0.003}$ |
| | AUC | $0.888_{0.003}$ | $0.845_{0.002}$ | $0.884_{0.003}$ | $0.868_{0.005}$ | $0.881_{0.002}$ | $0.884_{0.004}$ | $0.905_{0.004}$ | $0.913_{0.003}$ | $0.919_{0.002}$ |
| | 1-OE | $0.410_{0.007}$ | $0.295_{0.014}$ | $0.391_{0.009}$ | $0.311_{0.015}$ | $0.350_{0.009}$ | $0.468_{0.007}$ | $0.480_{0.002}$ | $0.489_{0.016}$ | $0.503_{0.020}$ |
| | 1-Cov | $0.723_{0.007}$ | $0.647_{0.005}$ | $0.718_{0.006}$ | $0.702_{0.008}$ | $0.725_{0.005}$ | $0.727_{0.011}$ | $0.777_{0.008}$ | $0.787_{0.009}$ | $0.804_{0.006}$ |
| Pascal07 | AP | $0.508_{0.005}$ | $0.471_{0.008}$ | $0.504_{0.005}$ | $0.437_{0.018}$ | $0.488_{0.003}$ | $0.505_{0.012}$ | $0.552_{0.003}$ | $0.555_{0.010}$ | $0.590_{0.008}$ |
| | 1-HL | $0.931_{0.001}$ | $0.928_{0.001}$ | $0.930_{0.000}$ | $0.882_{0.004}$ | $0.928_{0.001}$ | $0.929_{0.001}$ | $0.933_{0.007}$ | $0.931_{0.001}$ | $0.935_{0.001}$ |
| | 1-RL | $0.812_{0.004}$ | $0.761_{0.005}$ | $0.806_{0.003}$ | $0.736_{0.015}$ | $0.783_{0.001}$ | $0.783_{0.008}$ | $0.832_{0.007}$ | $0.830_{0.004}$ | $0.855_{0.004}$ |
| | AUC | $0.838_{0.003}$ | $0.779_{0.004}$ | $0.832_{0.002}$ | $0.767_{0.015}$ | $0.811_{0.001}$ | $0.809_{0.006}$ | $0.853_{0.003}$ | $0.850_{0.005}$ | $0.873_{0.004}$ |
| | 1-OE | $0.419_{0.008}$ | $0.420_{0.011}$ | $0.419_{0.008}$ | $0.362_{0.023}$ | $0.421_{0.006}$ | $0.427_{0.015}$ | $0.461_{0.007}$ | $0.464_{0.018}$ | $0.500_{0.010}$ |
| | 1-Cov | $0.759_{0.003}$ | $0.692_{0.004}$ | $0.751_{0.003}$ | $0.677_{0.015}$ | $0.727_{0.002}$ | $0.731_{0.006}$ | $0.785_{0.009}$ | $0.783_{0.006}$ | $0.805_{0.004}$ |
| ESPGame | AP | $0.289_{0.003}$ | $0.212_{0.002}$ | $0.285_{0.003}$ | $0.244_{0.005}$ | $0.246_{0.002}$ | $0.297_{0.002}$ | $0.312_{0.004}$ | $0.311_{0.004}$ | $0.319_{0.004}$ |
| | 1-HL | $0.983_{0.000}$ | $0.982_{0.000}$ | $0.983_{0.000}$ | $0.972_{0.000}$ | $0.983_{0.000}$ | $0.983_{0.000}$ | $0.983_{0.000}$ | $0.983_{0.000}$ | $0.983_{0.000}$ |
| | 1-RL | $0.832_{0.001}$ | $0.781_{0.001}$ | $0.829_{0.001}$ | $0.808_{0.002}$ | $0.818_{0.002}$ | $0.832_{0.001}$ | $0.847_{0.002}$ | $0.849_{0.002}$ | $0.857_{0.002}$ |
| | AUC | $0.836_{0.001}$ | $0.785_{0.001}$ | $0.833_{0.002}$ | $0.813_{0.002}$ | $0.824_{0.002}$ | $0.836_{0.001}$ | $0.852_{0.002}$ | $0.853_{0.002}$ | $0.861_{0.002}$ |
| | 1-OE | $0.396_{0.005}$ | $0.294_{0.006}$ | $0.389_{0.004}$ | $0.343_{0.013}$ | $0.339_{0.003}$ | $0.439_{0.007}$ | $0.461_{0.010}$ | $0.455_{0.007}$ | $0.460_{0.010}$ |
| | 1-Cov | $0.574_{0.004}$ | $0.488_{0.003}$ | $0.567_{0.005}$ | $0.548_{0.004}$ | $0.571_{0.003}$ | $0.593_{0.003}$ | $0.628_{0.005}$ | $0.628_{0.005}$ | $0.647_{0.004}$ |
| IAPRTC12 | AP | $0.305_{0.004}$ | $0.234_{0.003}$ | $0.304_{0.004}$ | $0.237_{0.003}$ | $0.261_{0.001}$ | $0.323_{0.001}$ | $0.339_{0.003}$ | $0.331_{0.006}$ | $0.351_{0.005}$ |
| | 1-HL | $0.981_{0.000}$ | $0.980_{0.000}$ | $0.981_{0.000}$ | $0.969_{0.000}$ | $0.980_{0.000}$ | $0.981_{0.000}$ | $0.980_{0.000}$ | $0.980_{0.000}$ | $0.981_{0.000}$ |
| | 1-RL | $0.862_{0.002}$ | $0.823_{0.002}$ | $0.861_{0.002}$ | $0.833_{0.002}$ | $0.848_{0.001}$ | $0.873_{0.001}$ | $0.886_{0.002}$ | $0.885_{0.003}$ | $0.895_{0.003}$ |
| | AUC | $0.864_{0.002}$ | $0.825_{0.001}$ | $0.863_{0.001}$ | $0.835_{0.001}$ | $0.850_{0.001}$ | $0.874_{0.000}$ | $0.888_{0.004}$ | $0.886_{0.002}$ | $0.896_{0.002}$ |
| | 1-OE | $0.432_{0.008}$ | $0.340_{0.006}$ | $0.429_{0.009}$ | $0.352_{0.008}$ | $0.390_{0.005}$ | $0.468_{0.002}$ | $0.463_{0.002}$ | $0.463_{0.009}$ | $0.486_{0.008}$ |
| | 1-Cov | $0.597_{0.004}$ | $0.529_{0.004}$ | $0.597_{0.004}$ | $0.564_{0.005}$ | $0.592_{0.004}$ | $0.649_{0.001}$ | $0.688_{0.007}$ | $0.675_{0.007}$ | $0.700_{0.006}$ |
| Mirflickr | AP | $0.570_{0.002}$ | $0.514_{0.006}$ | $0.553_{0.002}$ | $0.490_{0.012}$ | $0.551_{0.002}$ | $0.589_{0.005}$ | $0.611_{0.002}$ | $0.614_{0.004}$ | $0.633_{0.004}$ |
| | 1-HL | $0.886_{0.001}$ | $0.878_{0.001}$ | $0.885_{0.001}$ | $0.839_{0.002}$ | $0.882_{0.001}$ | $0.888_{0.002}$ | $0.891_{0.001}$ | $0.891_{0.001}$ | $0.895_{0.001}$ |
| | 1-RL | $0.856_{0.001}$ | $0.831_{0.003}$ | $0.856_{0.001}$ | $0.803_{0.008}$ | $0.844_{0.001}$ | $0.863_{0.004}$ | $0.875_{0.001}$ | $0.877_{0.002}$ | $0.888_{0.002}$ |
| | AUC | $0.846_{0.001}$ | $0.828_{0.003}$ | $0.844_{0.001}$ | $0.787_{0.012}$ | $0.837_{0.001}$ | $0.849_{0.004}$ | $0.862_{0.001}$ | $0.860_{0.003}$ | $0.874_{0.002}$ |
| | 1-OE | $0.631_{0.004}$ | $0.510_{0.008}$ | $0.607_{0.004}$ | $0.511_{0.022}$ | $0.585_{0.003}$ | $0.637_{0.007}$ | $0.662_{0.006}$ | $0.662_{0.008}$ | $0.683_{0.008}$ |
| | 1-Cov | $0.640_{0.001}$ | $0.604_{0.005}$ | $0.636_{0.001}$ | $0.572_{0.013}$ | $0.631_{0.002}$ | $0.652_{0.007}$ | $0.681_{0.003}$ | $0.678_{0.003}$ | $0.694_{0.003}$ |

## 4.2. Experimental Results and Analysis

In this section, we compare our method with other eight popular algorithms on the five datasets mentioned above and the experimental results under 50% missing-view rate and 50% missing-label rate of the six evaluation metrics are shown in Table 1, and we can have the following observations:

- Compare to other eight popular methods on the five datasets, our **COME** achieves the best performance on all metrics which fully verifies the effectiveness of our method on iM3C task.

- Compared to models that only consider single missing data, models that address the dual missing problem exhibit better performance and robustness. These findings offer valuable insights for guiding the design of models for iM3C tasks in the future.

To further investigate the impact of different missing view and missing label ratios on classification performance, we conducted tests on the Pascal07 dataset. The results are shown in Fig. 4. It can be observed that, under the same

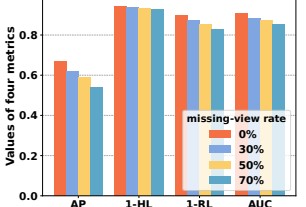
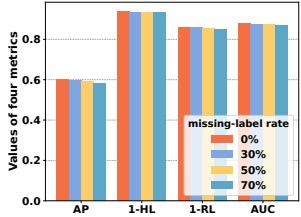

(a) various missing view rates   (b) various missing label rates

*Figure 4.* The four metrics value on Pascal07 dataset with (a) different missing-view rates and 50% missing-label rate and (b) 50% missing-view rate and different missing-label rates.

missing rate, view missing has a more severe impact on performance improvement compared to label missing.

To further validate the adaptability of our model, we conducted experiments using datasets without any missing views and labels. The results on the five dataset are illustrated in the form of a radar chart, as shown in Fig. 5 (refer to Appendix F for more results on other databases). It can be observed that our **COME** achieves superior performance across six metrics compared to other methods, including

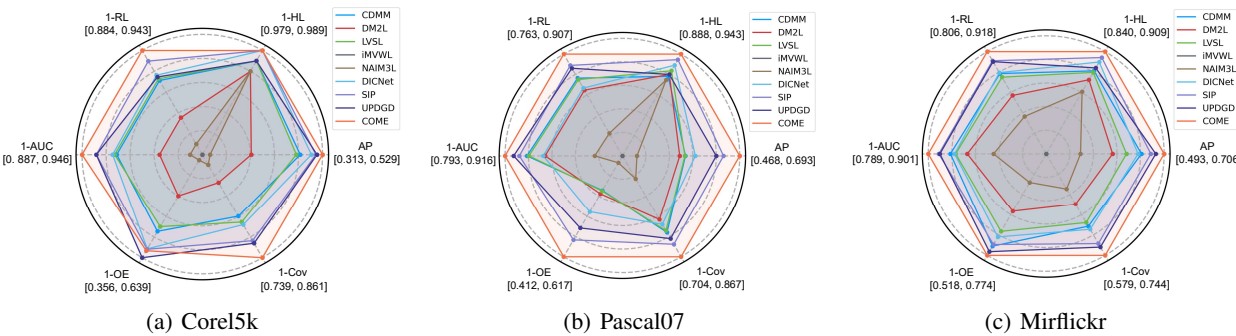

(a) Corel5k       (b) Pascal07       (c) Mirflickr

*Figure 5.* The experimental results of nine methods on the complete Corel5k, Pascal07, and Mirflickr dataset. In this visualization, the worst results are positioned at the center, while the best results correspond to the vertices, based on six evaluation metrics.

those designed for ideal complete cases. This demonstrates the excellent generalization capability of our model.

### 4.3. Shared Information Compression v.s. Information Reconstruction

As we know, contrastive learning is a typical information compression strategy, while our proposed mutual information enhancement strategy based on MoE emphasizes the maintenance of effective shared information. That is to say, the learned shared information remains relevant to the original data to prevent information collapse. In this section, we study the equilibrium relationship between $\mathcal{L}_{RE1} + \mathcal{L}_{RE2}$ and $\mathcal{L}_{CTR}$ by changing the value of parameter $\beta$. As shown in Fig. 6, when the value of $\beta$ is 0.1 and 1 for Corel5k and Pascal07 datasets, respectively, information compression and effective information reconstruction reach a balanced state, and the model achieves the optimal performance. If we continue to increase $\beta$, we can find that AP value decreases sharply, which indicates that excessive information compression caused by contrastive learning makes it difficult for the model to learn meaningful information for downstream prediction.

*Table 2.* The ablation experimental results on the Corel5K and Pascal07 datasets, and all datasets are with 50% missing views, 50% missing labels and 70% training samples.

| Backbone | $\mathcal{L}_{RE1}$ | $\mathcal{L}_{RE2}$ | $\mathcal{L}_{CE}$ | Corel5k | | Pascal07 | |
|---|---|---|---|---|---|---|---|
| | | | | AP | AUC | AP | AUC |
| ✓ | | | | 0.390 | 0.883 | 0.535 | 0.846 |
| ✓ | ✓ | | | 0.399 | 0.908 | 0.581 | 0.868 |
| ✓ | | ✓ | | 0.402 | 0.910 | 0.562 | 0.862 |
| ✓ | | | ✓ | 0.390 | 0.903 | 0.538 | 0.847 |
| ✓ | ✓ | ✓ | | 0.427 | 0.918 | 0.586 | 0.872 |
| ✓ | ✓ | | ✓ | 0.406 | 0.912 | 0.584 | 0.871 |
| ✓ | | ✓ | ✓ | 0.404 | 0.910 | 0.565 | 0.863 |
| ✓ | ✓ | ✓ | ✓ | 0.432 | 0.919 | 0.590 | 0.873 |

### 4.4. Ablation Study

The ablation experiments are conducted on Corel5k and Pascal07 datasets, in which the missing-view rate and missing-label rate are both 50%. Our objective function consists of two parts: the minimal sufficient shared representation learning loss $\mathcal{L}_{MSR}$ and classification loss $\mathcal{L}_{CE}$. Next, we will decompose the loss and conduct ablation experiments from three perspectives: examining the effects of intra-view mutual information enhancement, multi-view mutual information enhancement, and pseudo-labels. The corresponding loss of these three parts are $\mathcal{L}_{RE1}$, $\mathcal{L}_{RE2}$, and $\mathcal{L}_{CE}$. It is worth noting that when pseudo-labels are not used, the supervisory information for the model comes from the original labels that are not missing. The results presented in Table 2, and it can be observed that each component of our **COME** is crucial and contributes positively. It is worth noting that the multi-view mutual information enhancement module plays a crucial role in improving performance.

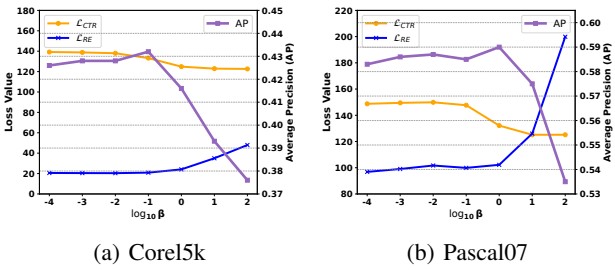

(a) Corel5k       (b) Pascal07

*Figure 6.* Information balance study: AP and losses v.s. $\beta$ on Corel5k and Pascal07 datasets with 50% view and label missing rates.

## 5. Conclusion

In this paper, we assume that the consistent semantic information shared among views is sufficient for downstream tasks. Based on this, we propose a compact semantic learning framework, named **COME**, for iM3C task. Our moti-

vation to this issue consists of two parts: compact shared representation learning and soft pseudo-label generation. On the one hand, we alleviate the limitations of contrastive learning in handling incomplete data by maximizing mutual information between original data and cross-view joint representation, striving to learn the compact shared representation. On the other hand, we propose a dual-branch soft pseudo-label cross-imputation strategy to mitigate the impact of missing supervisory information. Finally, we conducted extensive experiments to validate the effectiveness of our method and its strong generalization capability.

## Acknowledgments

This work was supported in part by Shenzhen Science and Technology Program under Grant No. JCYJ20240813105135047; in part by Guangdong Basic and Applied Basic Research Foundation under Grant 2024A1515030213; and in part by National Natural Science Foundation of China under Grant 62372136.

## Impact Statement

This paper presents work whose goal is to advance the field of Machine Learning. There are many potential societal consequences of our work, none which we feel must be specifically highlighted here.

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

# A. Complete Derivation of Shared Representation Learning

Based on the analysis in Section 3.2, we propose to maximize the mutual information between the original data and the joint representation:

$$\max I(\mathbf{X}^{\mathcal{V}}; \mathbf{z}) \tag{14}$$

For mutual information $I(\mathbf{X}^{\mathcal{V}}; \mathbf{z})$ in Eq. (14), we can obtain the variational lower bound as follows:

$$
\begin{aligned}
&I(\mathbf{X}^{\mathcal{V}}; \mathbf{z}) \\
&= \int \int p(\mathbf{X}^{\mathcal{V}}, \mathbf{z}) \log \frac{p(\mathbf{X}^{\mathcal{V}}|\mathbf{z})}{p(\mathbf{X}^{\mathcal{V}})} d\mathbf{X}^{\mathcal{V}} d\mathbf{z} \\
&\geq \int p(\mathbf{X}^{\mathcal{V}}) \int p(\mathbf{z}|\mathbf{X}^{\mathcal{V}}) \log p(\mathbf{X}^{\mathcal{V}}|\mathbf{z}) d\mathbf{X}^{\mathcal{V}} d\mathbf{z} \\
&= \int p(\mathbf{X}^{\mathcal{V}}) \int p(\mathbf{z}|\mathbf{X}^{\mathcal{V}}) \log q(\mathbf{X}^{\mathcal{V}}|\mathbf{z}) d\mathbf{X}^{\mathcal{V}} d\mathbf{z} + \\
&\quad \int p(\mathbf{X}^{\mathcal{V}}) \int p(\mathbf{z}|\mathbf{X}^{\mathcal{V}}) \log \frac{p(\mathbf{X}^{\mathcal{V}}|\mathbf{z})}{q(\mathbf{X}^{\mathcal{V}}|\mathbf{z})} d\mathbf{X}^{\mathcal{V}} d\mathbf{z} \\
&\geq \mathbb{E}_{\mathbf{X}^{\mathcal{V}} \sim p(\mathbf{X}^{\mathcal{V}})} \left[ \int p(\mathbf{z}|\mathbf{X}^{\mathcal{V}}) \log q(\mathbf{X}^{\mathcal{V}}|\mathbf{z}) d\mathbf{z} \right]
\end{aligned} \tag{15}
$$

Similar to previous work(Shi et al., 2019), we propose to factorise the joint variational posterior as a combination of unimodal posteriors, using a Mixture Of Experts (MOE) to model $p(\mathbf{z}|\mathbf{X}^{\mathcal{V}})$, therefore we have:

$$p(\mathbf{z}|\mathbf{X}^{\mathcal{V}}) = \sum_{v \in \mathcal{V}} \frac{1}{|\mathcal{V}|} p(\mathbf{z}^{(v)}|\mathbf{x}^{(v)}) \tag{16}$$

Combined with Eq. (16), we can rewrite the variational lower bound in Eq. (15) as follows:

$$
\begin{aligned}
&\mathbb{E}_{\mathbf{X}^{\mathcal{V}} \sim p(\mathbf{X}^{\mathcal{V}})} \left[ \int p(\mathbf{z}|\mathbf{X}^{\mathcal{V}}) \log q(\mathbf{X}^{\mathcal{V}}|\mathbf{z}) d\mathbf{z} \right] \\
=& \mathbb{E}_{\mathbf{X}^{\mathcal{V}} \sim p(\mathbf{X}^{\mathcal{V}})} \left[ \int p(\mathbf{z}|\mathbf{X}^{\mathcal{V}}) \log \prod_{v \in \mathcal{V}} q(\mathbf{x}^{(v)}|\mathbf{z}) d\mathbf{z} \right] \\
=& \mathbb{E}_{\mathbf{X}^{\mathcal{V}} \sim p(\mathbf{X}^{\mathcal{V}})} \left[ \int \left( \sum_{v \in \mathcal{V}} \frac{1}{|\mathcal{V}|} p(\mathbf{z}^{(v)}|\mathbf{x}^{(v)}) \right) \log \prod_{v \in \mathcal{V}} q(\mathbf{x}^{(v)}|\mathbf{z}) d\mathbf{z} \right] \\
=& \frac{1}{|\mathcal{V}|} \sum_{v \in \mathcal{V}} \mathbb{E}_{\mathbf{x}^{(v)} \sim p(\mathbf{x}^{(v)})} \left[ \int p(\mathbf{z}|\mathbf{x}^{(v)}) \log q(\mathbf{x}^{(v)}|\mathbf{z}) d\mathbf{z} \right] \\
&+ \frac{1}{|\mathcal{V}|} \sum_{v \in \mathcal{V}} \sum_{u \in \mathcal{V}, u \neq v} \mathbb{E}_{\mathbf{x}^{(v)} \sim p(\mathbf{x}^{(v)})} \left[ \int p(\mathbf{z}|\mathbf{x}^{(v)}) \log q(\mathbf{x}^{(u)}|\mathbf{z}) d\mathbf{z} \right]
\end{aligned} \tag{17}
$$

For better clarity, we will use two views as an example for the subsequent derivation. We have the joint posterior probability $p(\mathbf{z}|\{\mathbf{x}^{(1)}, \mathbf{x}^{(2)}\})$. And the variational lower bound can be rewrite as follows:

$$
\begin{aligned}
&\mathbb{E}_{\mathbf{X}^{\mathcal{V}} \sim p(\mathbf{X}^{\mathcal{V}})} \left[ \int p(\mathbf{z}|\mathbf{X}^{\mathcal{V}}) \log q(\mathbf{X}^{\mathcal{V}}|\mathbf{z}) d\mathbf{z} \right] \\
=& \frac{1}{2} \mathbb{E}_{\mathbf{X}^{\mathcal{V}} \sim p(\mathbf{X}^{\mathcal{V}})} \left[ \int (p(\mathbf{z}|\mathbf{x}^{(1)}) + (p(\mathbf{z}|\mathbf{x}^{(2)}))) \log q(\{\mathbf{x}^{(1)}, \mathbf{x}^{(2)}\}|\mathbf{z}) d\mathbf{z} \right] \\
=& \frac{1}{2} \mathbb{E}_{\mathbf{X}^{\mathcal{V}} \sim p(\mathbf{X}^{\mathcal{V}})} \left[ \int (p(\mathbf{z}|\mathbf{x}^{(1)}) + (p(\mathbf{z}|\mathbf{x}^{(2)}))) \log[q(\mathbf{x}^{(1)}|\mathbf{z}) q(\mathbf{x}^{(2)}|\mathbf{z})] d\mathbf{z} \right] \\
=& \frac{1}{2} \mathbb{E}_{\mathbf{X}^{\mathcal{V}} \sim p(\mathbf{X}^{\mathcal{V}})} \left[ \int p(\mathbf{z}|\mathbf{x}^{(1)}) \log q(\mathbf{x}^{(1)}|\mathbf{z}) d\mathbf{z} + \int p(\mathbf{z}|\mathbf{x}^{(1)}) \log q(\mathbf{x}^{(2)}|\mathbf{z}) d\mathbf{z} \right. \\
&\left. + \int p(\mathbf{z}|\mathbf{x}^{(2)}) \log q(\mathbf{x}^{(1)}|\mathbf{z}) d\mathbf{z} + \int p(\mathbf{z}|\mathbf{x}^{(2)}) \log q(\mathbf{x}^{(2)}|\mathbf{z}) d\mathbf{z} \right]
\end{aligned} \tag{18}
$$

*Table 3.* Detailed information about five multi-view multi-label datasets in our expriments.

| Dataset | # Sample | # Label | # View | # Label/#Sample |
|---------|----------|---------|--------|-----------------|
| Corel5k | 4999 | 260 | 6 | 3.40 |
| IAPRTC12 | 19627 | 291 | 6 | 5.72 |
| ESPGame | 20770 | 268 | 6 | 4.69 |
| Pascal07 | 9963 | 20 | 6 | 1.47 |
| Mirflickr | 25000 | 38 | 6 | 4.72 |

*Table 4.* The concise summary of comparative methods, highlighting their capabilities: The attributes 'Multi-view' and 'Incomplete-view' indicate a method's ability to handle multi-view and incomplete multi-view scenarios, respectively, while the attribute 'Missing-label' reflects its capability to address incomplete multi-label classification tasks.

| Method | Sources | Multi-view | Incomplete-view | Missing-label |
|--------|---------|------------|-----------------|---------------|
| CDMM | KBS'20 | ✓ | × | × |
| DM2L | PR'21 | × | × | ✓ |
| LVSL | TMM'22 | ✓ | × | × |
| iMVWL | IJCAI'18 | ✓ | ✓ | ✓ |
| NAIM3L | TPAMI'21 | ✓ | ✓ | ✓ |
| DICNet | AAAI'23 | ✓ | ✓ | ✓ |
| UPDGD-Net | ACM MM'24 | ✓ | ✓ | ✓ |
| SIP | ICML'24 | ✓ | ✓ | ✓ |

## B. Statistics for Five Datasets and Data Masking Schemes

In this section, we provide detailed information about the five datasets used in the experiments, as summarized in Table 3. To emulate real-world scenarios involving missing views and partial labels, we generate incomplete multi-view partial multi-label data through the following steps: (1) Missing views: For each view, 50% of the samples are randomly masked, while ensuring that each sample retains at least one available view. (2) Missing labels: For each category, 50% of the positive and negative labels are randomly removed. (3) Dataset Splitting: Subsequently, 70% of the resulting samples are randomly selected as the training set.

## C. Statistics for Eight Competitors

In this section, we present detailed information about the eight comparison methods, as summarized in Table 4. Specifically, DM2L can only deal with single-view partial multi-label case, so we record results of each view and select the best results. CDMM and LVSL cannot deal with missing view, so we handle missing instances by imputing them with the mean values of the corresponding view's available instances. And the rest five methods have the capability to handle both incomplete views and partial labels.

## D. Sensitivity Studies of Hyper-parameters

We discussed enhancing the performance of learned representations through two approaches: increasing intra-modal mutual information and inter-modal mutual information. Additionally, we use two trade-off coefficients $\lambda_1$ and $\lambda_2$ to maintain a balance. We explore the sensitivity of these two hyperparameters on different datasets with 50% missing view, 50% missing label and 70% training samples. As shown in Fig. 7, based on experiments, it can be observed that the model's performance is not significantly affected by changes in the values of $\lambda_1$ and $\lambda_2$, highlighting the model's inherent robustness.

## E. Label Count Distribution across Samples in Other Datasets.

In this section, we present the label count distribution information for other datasets. This further demonstrates that the number of labels corresponding to each sample is not fixed, making Top-1 or Top-k pseudo-label generation strategies difficult to apply.

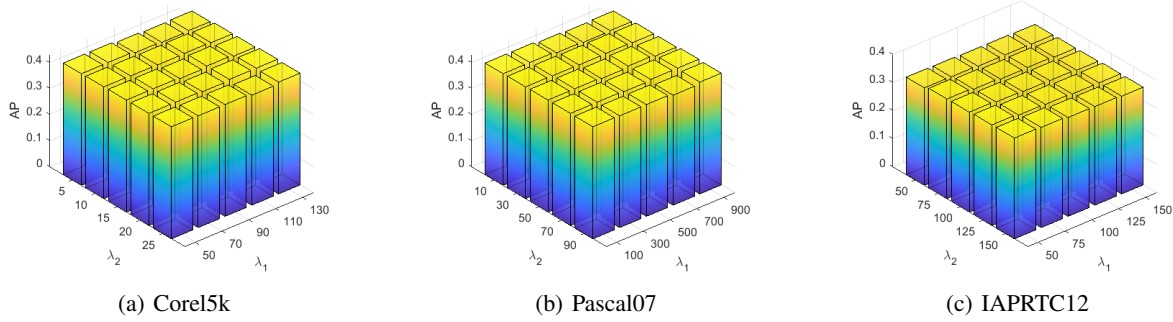

*Figure 7.* Hyper-parameters analysis regarding $\lambda_1$ and $\lambda_2$ on the Corel5k and Pascal07 databases; All datasets are setting to 50% views and labels missing rates.

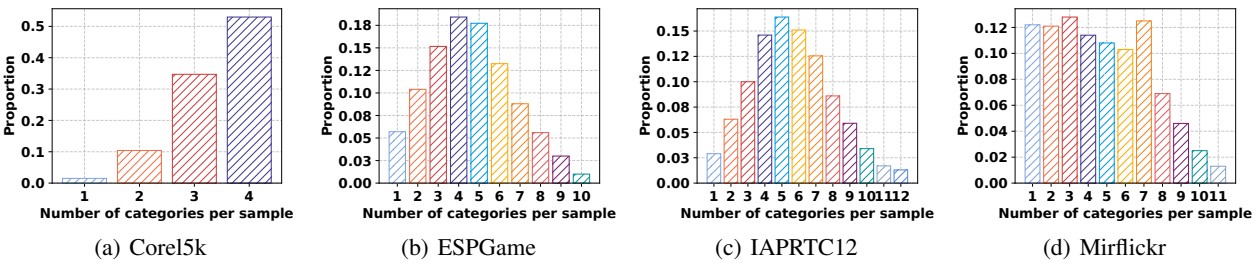

*Figure 8.* Label count distribution across samples in other four datasets.

# F. Extra Experimental Results on Five Full Datasets

In this section, we present the value of six metrics of nine methods in four datasets without any missing views or labels in the form of radar charts, as shown in Fig. 9.

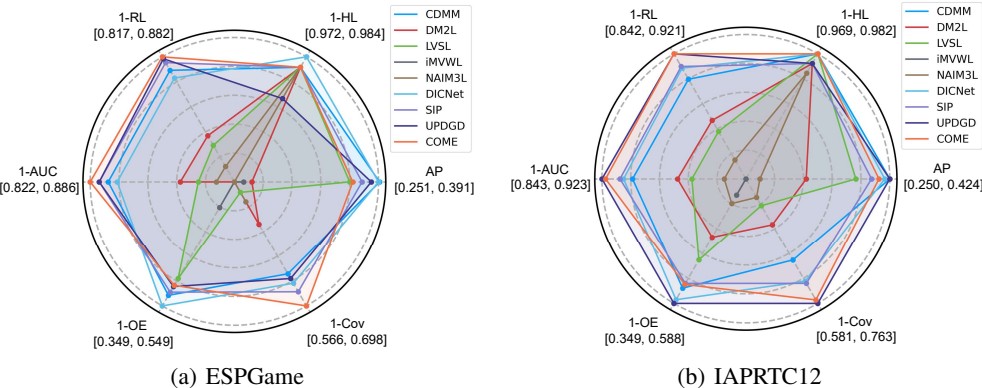

*Figure 9.* The experimental results on the rest two dataset without any missing views and labels. The worst results are indicated at the center of the radar chart, while the best results are represented by the vertices, based on six evaluation metrics.

