# OpenReview forum: "Learning Compact Semantic Information for Incomplete Multi-View Missing Multi-Label Classification"
_ICML.cc/2025/Conference — ICML 2025 poster_

### Official Review · Reviewer_8PHw · 2025-03-06

**Overall Recommendation:** 4

**Summary:**

The paper develops a model named COME for incomplete multi-view missing multi-label classification tasks. Unlike most existing methods, the approach aims to learn compact semantic information by minimizing task-independent redundant information. Additionally, a dual-branch soft pseudo-label generation strategy is introduced in the model to alleviate the negative impact of missing supervisory information.

**Claims And Evidence:**

The claims are clear and convincing. Main claim: the proposed multi-view semantic consistency enhancement strategy learns compact multi-view shared information, which effectively alleviates the performance degradation caused by incomplete contrastive learning.

**Essential References Not Discussed:**

The literature citations of the article are reasonable, and the latest relevant theories and specific methods are introduced in detail.

**Experimental Designs Or Analyses:**

A large number of experiments are carried out on five datasets. The experimental settings are reasonable, and the experimental analysis is sufficient.

**Methods And Evaluation Criteria:**

The authors used six evaluation metrics commonly used in multi-label classification, which are rational and common.

**Other Comments Or Suggestions:**

I noticed that in the experimental results on the complete dataset in Figure 8, there are only results of UPDGD on the Corel5k and Pascal07 datasets. Could you provide more results on other datasets?

**Other Strengths And Weaknesses:**

Strengths:
1. The paper is well-written, with explicit problem definition and clear description. It’s readily understandable even without prior knowledge.
2. The paper provides comprehensive experimental results to validate the effectiveness and robustness of the proposed methods.
3. The setting of double incompleteness in views and labels is novel, and the proposed pseudo-label strategy alleviates the problem of missing labels.

Weaknesses:
1. Some minor mistakes should be revised, the reference of equation 14 in line 205 is incorrect and it should be capitalized in line 374: “In table 1”.
2. The code is not provided. To improve reproducibility, implementation details and code should be provided.

**Questions For Authors:**

1.	Is the data still incomplete in the inference phase? I know that views and labels are not complete in the training, so is this still the case when inferencing?
2.	In Figure 2(b), is this label distribution common? Are the patterns found in a single dataset universal?
3.	From Figure 4, why is the negative impact of missing labels smaller than that of missing views? More discussions should be added.

**Relation To Broader Scientific Literature:**

This paper studies the inadequacy of contrastive learning in dealing with incomplete multi-view data, which is worth further exploration. Unlike most existing methods, the proposed method aims to learn compact semantic information by minimizing task-independent redundant information.

**Theoretical Claims:**

The adequacy hypothesis of multiple views shared information proposed in the paper is reasonable.

---

> ### Author Rebuttal · Authors · 2025-03-31
>
> Thanks for your constructive reviews and suggestions. Below, we will address each of your questions.
> > Q1: Some minor mistakes should be revised, the reference of equation 14 in line 205 is incorrect and it should be capitalized in line 374: “In table 1”.
>
> Thanks for your correction. We have corrected the incorrect reference to Eq. (14) in line 205, capitalized “Table 1” in line 374, and addressed other minor errors throughout the manuscript to ensure consistency with academic standards.
>
> > Q2: The code is not provided. To improve reproducibility, implementation details and code should be provided.
>
> Thanks for your suggestion. The code will be made publicly available upon acceptance to support reproducibility.
>
> > Q3: I noticed that in the experimental results on the complete dataset in Figure 8, there are only results of UPDGD on the Corel5k and Pascal07 datasets. Could you provide more results on other datasets?
>
> We sincerely apologize for this and we have provided the complete experimental results on the five datasets without any missing views and labels in Figure 1 of the PDF document (provided via the anonymized link). It can be observed that our Model (COME) achieves superior performance across six metrics compared to other methods on most datasets.
>
> > Q4: Is the data still incomplete in the inference phase? I know that views and labels are not complete in the training, so is this still the case when inferencing?
>
> We still maintain the view’s incompleteness in the inference phase, but we do not use any missing settings for labels as the evaluation criteria for real performance.
>
> > Q5: In Figure 2(b), is this label distribution common? Are the patterns found in a single dataset universal?
>
> Thanks to your questions. The label distribution of the other datasets is summarized in Figure 2 of the PDF document. We observed that the number of labels per sample varies across datasets and within individual datasets. This inherent variability makes the Top-K pseudo-labeling strategy [1] unsuitable for such scenarios.
>
> > Q6: From Figure 4, why is the negative impact of missing labels smaller than that of missing views? More discussions should be added.
>
> Thank you for your discussion. Intuitively, in the fusion stage of multi-view representations using a Mixture of Experts (MoE), the learned weights reflect the relative importance of different view representations within the fused representation. The absence of key view representations causes rapid degradation in the quality of multi-view joint representations, which poses significant challenges to multi-label classifiers. Additionally, we introduce a dual-branch soft pseudo-label imputation strategy to mitigate the multi-label missing problem. We present the ablation results of the “dual-branch soft pseudo-label imputation” strategy under varying missing label rates.  The results of COME without dual-branch soft pseudo-label imputation strategy are summarized in the following table:
> | label-missing rate | AP    | 1-HL  | 1-RL  | AUC   |
> |--------------------|-------|-------|-------|-------|
> | 0%                | 0.604 | 0.937 | 0.862 | 0.879 |
> | 30%               | 0.594 | 0.935 | 0.857 | 0.875 |
> | 50%               | 0.586 | 0.935 | 0.854 | 0.872 |
> | 70%               | 0.571 | 0.934 | 0.843 | 0.864 |
>
> And the results of COME with dual-branch soft pseudo-label imputation strategy are summarized in the following table:
> | label-missing rate | AP    | 1-HL  | 1-RL  | AUC   |
> |--------------------|-------|-------|-------|-------|
> | 0%                | 0.602 | 0.937 | 0.859 | 0.878 |
> | 30%               | 0.596 | 0.936 | 0.859 | 0.876 |
> | 50%               | 0.590 | 0.935 | 0.855 | 0.873 |
> | 70%               | 0.580 | 0.933 | 0.852 | 0.870 |
>
> For clarity, the results are presented in Figure 3 of the PDF. The experimental results demonstrate that the proposed strategy effectively mitigates the negative impact of missing labels, particularly under a high missing rate of 70%.
>
> References:
> [1] Class-Distribution-Aware Pseudo-Labeling for Semi-Supervised Multi-Label Learning. NeurIPS 2023.
>
> Anonymous link:
> https://anonymous.4open.science/r/6513/rebuttal.pdf

---

> > ### Comment · Reviewer_8PHw · 2025-04-07
> >
> > Thanks for the reply. I have no further questions and decide to keep my rating.

---

> > > ### Author Response · Authors · 2025-04-07
> > >
> > > We appreciate your careful review and valuable comments on the manuscript. We attach great importance to your feedback and incorporate your suggestions to improve the final version of the paper.

---

### Official Review · Reviewer_f1Pg · 2025-03-08

**Overall Recommendation:** 4

**Summary:**

The method COME is developed to address the missing data in multi-view multi-label classification tasks, which pursues the maximization of cross-view information to compress the irrelevant information and develops a pseudo-label filling strategy to handle the unavailable labels. Besides, the authors claim that missing data leads to insufficient contrastive learning and then build an information theory-based model to handle it.

**Claims And Evidence:**

1. Claim: missing data results in insufficient contrastive learning. Evidence: shown in Figure 1.
2. Claim: all the task-relevant information is contained by multi-view shared information. Evidence: Assumption 2.1 and Proposition 2.2.

**Essential References Not Discussed:**

No

**Experimental Designs Or Analyses:**

The authors compared eight methods across five datasets in both incomplete and complete cases. Ablation experiments and parameter analysis experiments were conducted. It would be beneficial for the authors to clarify through experiments why dual-branch model is preferred compared to a single-branch structure.

**Methods And Evaluation Criteria:**

The authors propose a compact semantic learning framework, named COME, for iM3C task. Six metrics such as AUC, AP, OE and so on are used to evaluate the performance of the method.

**Other Comments Or Suggestions:**

1. The cross-view reconstruction in Eq.(8) does not use the joint representation z. Could you explain it? Additionally, is the joint representation z utilized in the classification tasks? I think the authors should provide a detailed clarification on this.
2. In Figure 7, the hyperparameters $\lambda_1$ and $\lambda_2$ appear to tiny influence on the experimental outcomes. Could the authors elaborate on the empirical or theoretical rationale behind this observed insensitivity?

**Other Strengths And Weaknesses:**

**Strengths**
1. Understanding the performance degradation mechanisms of contrastive learning under incomplete multi-view scenarios remains an open challenge, with significant implications for robust representation learning. The authors proposed an effective solution to this problem and I think this framework has the potential to be extended to other multi-view learning tasks.
2. Overall, the experimental results presented in this paper are rich and convincing.
3. The description of the method is clear and understandable, which I think will be helpful for readers to replicate the results.
4. The authors study the complexity of the multi-view multi-label classification problem from various aspects, including multi-view representation learning and multi-label classification, from feature extraction to pseudo-label generation, showing a large workload.

**Weaknesses**
1. Some details need to be improved, such as the typo “we suppose all the task-relevant information contained by multi-view shared information.” in line 97. I suggest the authors to polish it to help improve the fluency.
2. Equations in the Appendix should not appear in the main text commonly, such as Eq. (14).
3. The authors claim that in pseudo-label generation, the use of hard pseudo-labels may lead to error accumulation and over-fitting, but there is no relevant comparative experiment in the paper. Please present the comparative experiments using hard labels and soft pseudo-labels.

**Questions For Authors:**

1. The difference in Figure 4(b) is very small. Is this caused by using pseudo labels? How does the model perform at different missing rates if the pseudo-label module is disabled?
2. Figure 3 does not seem to be mentioned in the text. The font styling in Figure 6 is inconsistent.

**Relation To Broader Scientific Literature:**

In contrast to prior studies, the manuscript explores how contrastive learning degenerates in multi-view multi-label classification when both view and label-missing coexist. By learning compact representations, the degeneration issue is alleviated, thus enabling contrastive learning to remain robust even in the case of view missing. The authors proposed a novel dual-branch architecture to generate soft pseudo-labels, effectively addressing the label missing problem.

**Theoretical Claims:**

Assumption 2.1 and Proposition 2.2 provide the basic theoretical claims for the method, which is solid and reliable according to existing methods.

---

> ### Author Rebuttal · Authors · 2025-03-31
>
> We greatly appreciate your thoughtful and detailed feedback, and we will address your questions one by one.
> > Q1: It would be beneficial for the authors to clarify through experiments why dual-branch model is preferred compared to a single-branch structure.
>
> Thank you for your suggestions. We carried out experiments on the Pascal07 dataset to investigate the differences between dual-branch and single-branch architectures. The results are summarized in the following table:
> | Model                  | AP     | 1-HL   | 1-RL   | AUC    |
> |------------------------|--------|--------|--------|--------|
> | COME(dual-branch)      | 0.590  | 0.935  | 0.855  | 0.873  |
> | COME(single-branch)    | 0.586  | 0.935  | 0.852  | 0.872  |
>
> The experimental results demonstrate that the dual-branch architecture exhibits superior performance compared to its single-branch counterpart.
>
> > Q2: Some details need to be improved, such as the typo “we suppose all the task-relevant information contained by multi-view shared information.” in line 97. I suggest the authors to polish it to help improve the fluency.
>
> Thank you for your suggestion. In line 97, the original sentence has been revised to the following in the manuscript: “We suppose that all the task-relevant information is contained in the multi-view shared representation.”.
>
> > Q3: Equations in the Appendix should not appear in the main text commonly, such as Eq. (14).
>
> I am sorry for the serious typos. The corresponding equation is Eq. (6) rather than Eq. (14). We have updated this in the revised  manuscript.
>
> > Q4: Please present the comparative experiments using hard labels and soft pseudo-labels.
>
> Thank you for your suggestion. We compared ​the soft pseudo-labeling strategy with ​the hard pseudo-labeling strategy and summarize the results in the follwing table:
> | Model                        | dataset   | AP    | 1-HL  | 1-RL  | AUC   |
> |------------------------------|-----------|-------|-------|-------|-------|
> | COME with hard pseudo-labeling | Corel5k  | 0.425 | 0.988 | 0.916 | 0.918 |
> | COME with hard pseudo-labeling | Pascal07 | 0.585 | 0.934 | 0.852 | 0.872 |
> | COME with soft pseudo-labeling | Corel5k  | 0.432 | 0.988 | 0.917 | 0.920 |
> | COME with soft pseudo-labeling | Pascal07 | 0.590 | 0.935 | 0.854 | 0.873 |
>
> The experimental results demonstrate that the soft pseudo-labeling strategy exhibits superior performance compared to the hard pseudo-labeling approach.
>
> > Q5: The cross-view reconstruction in Eq.(8) does not use the joint representation z. Could you explain it? Additionally, is the joint representation z utilized in the classification tasks? I think the authors should provide a detailed clarification on this.
>
> Thank you for your question. Combining the Mixture of Experts (MOE) with Eq. (15), we derive Eq. (17). This implies that the mutual information between a given view and others can be enhanced through cross-view representation reconstruction, thereby improving semantic consistency. Moreover, the joint representation z has indeed been utilized in multi-label classification tasks, and we have supplemented this in the revised manuscript.
>
> > Q6: In Figure 7, the hyperparameters and appear to tiny influence on the experimental outcomes. Could the authors elaborate on the empirical or theoretical rationale behind this observed insensitivity?
>
> Thank you for your question. We conducted sensitivity analysis experiments to investigate the influence of $\lambda_1$ and $\lambda_2$ on five datasets by varying their values. Subsequently, we identified hyperparameter configurations that ensured stable model performance and refined the intervals for detailed analysis. These results are visualized in Figure 4 of the PDF document (available via the anonymized link). The experiments show that the model maintains stable performance across a wide range of  $\lambda_1$ and $\lambda_2$, demonstrating high robustness to these hyperparameters.
>
> > Q7: The difference in Figure 4(b) is very small. Is this caused by using pseudo labels? How does the model perform at different missing rates if the pseudo-label module is disabled?
>
> Thank you for your valuable discussion. We conducted ablation experiments on the pseudo-labeling strategy under varying label missing rates. The performance of COME without this strategy is summarized in Table 1 of the provided PDF document. And the results of COME with the proposed strategy are presented in Table 2 of the PDF. For a clear observation, we show the results in Figure 3 of the PDF. The experimental results demonstrate that the proposed strategy mitigates the negative effects caused by label missing.
>
> > Q8: Figure 3 does not seem to be mentioned in the text. The font styling in Figure 6 is inconsistent.
>
> Thanks to the suggestion. The manuscript will be revised to include extended annotations for Figure 3 and to ensure font style consistency in Figure 6.
>
> Anonymous link:
> https://anonymous.4open.science/r/6513/rebuttal.pdf

---

> > ### Comment · Reviewer_f1Pg · 2025-04-03
> >
> > Thanks for your responses. They have addressed my concerns.

---

> > > ### Author Response · Authors · 2025-04-07
> > >
> > > We thank you for your thorough review of our paper and for providing constructive feedback that has significantly contributed to its improvement. Your insights have been invaluable in helping us refine our work.

---

### Official Review · Reviewer_oFHQ · 2025-03-11

**Overall Recommendation:** 3

**Summary:**

The authors delve into the study of incomplete Multi-view Missing Multi-Label Classification (iM3C) and aim to address the inadequacy of contrastive learning in dealing with incomplete multi-view data and the negative impact of missing labels. To tackle this problem, they propose a consistent semantic representation learning framework named COME. Firstly, COME learns the minimum sufficient representation by maximizing the mutual information across views. Secondly, it employs a dual-branch soft pseudo-label cross-imputation strategy to mitigate the negative impact of missing supervisory information. To verify the effectiveness of COME, the authors conduct experiments across various datasets and missing settings.

**Claims And Evidence:**

They claim that a model is proposed to solve the double missing problem of view and label, and experiments confirm the effectiveness of the proposed model.

**Essential References Not Discussed:**

There are no Not Discussed Essential References have been found.

**Experimental Designs Or Analyses:**

The experimental design is detailed, and the experiments are conducted separately under different missing rates.

**Methods And Evaluation Criteria:**

The proposed method has obvious pertinence to the iM3C problem, and the metrics used in the paper are also used in many related literatures.

**Other Comments Or Suggestions:**

See weakness.

**Other Strengths And Weaknesses:**

Strengths:
1) The mutual information enhancement strategy effectively compresses redundant information while retaining task-relevant shared semantic information, improving representation learning for incomplete data.
2)	The dual-branch soft pseudo-label generation innovatively reduces the negative impact of missing labels, avoiding error amplification compared to hard pseudo-labeling methods.
3)	The paper is well-written and has high readability.

Weaknesses:
1) Three important hyperparameters in equation 13, \lambda_1,  \lambda_2,   \beta, need the subsequent experiment to analyze their impact.
2)	The authors should check the full text carefully for grammatical errors, such as ‘consitent’ in line 107, “random” in line 116, to meet the high quality requirements of ICML.
3)	In line 112, “As shown in Fig. 4”, what does that mean? You may want to say “Fig. 1”.
4)	Subfigure needs captions as well, such as that in Fig. 2.
5) In line 319, “further technical” should be revised to “and further technical”.

**Questions For Authors:**

1) Why did the author use two separate models for pseudo-tag generation and padding?
2) In Eq. 11, whether the selection of threshold value is empirical or not?

**Relation To Broader Scientific Literature:**

This article provides a new approach to solving the problem of insufficient contrastive learning from the perspective of information theory. Based on existing works, it further studies the beneficial effects of mutual information maximization on multi-view learning.

**Theoretical Claims:**

The authors suppose all the task-relevant information contained by multi-view shared information, which is intuitive and used in previous works.

---

> ### Author Rebuttal · Authors · 2025-03-31
>
> Thank you for your valuable review. We will address your questions one by one.
> > Q1: Three important hyperparameters in equation 13, $\lambda_1$, $\lambda_2$, $\beta$, need the subsequent experiment to analyze their impact.
>
> Thank you for the suggestion. In Appendix D, we investigate the impacts of hyperparameters $\lambda_1$ and $\lambda_2$ across three datasets. Experimental results indicate that our method is robust to hyperparameter variations. For visual clarity, Figure 4 in the PDF (provided via the anonymized link) illustrates heatmaps of the average precision (AP) on two datasets. Additionally, in Section 4.3, we analyze the hyperparameter $\beta$, which balances view-shared and view-specific information. When the $\beta$ is too small, the model tends to focus excessively on view-specific representations, hindering its ability to learn effective shared representations. Conversely, when $\beta$ is too large, the model overemphasizes the consistency of shared representations, thereby compressing an excessive amount of information.
>
> > Q2: The authors should check the full text carefully for grammatical errors, such as ‘consitent’ in line 107, “random” in line 116, to meet the high quality requirements of ICML.
>
> Thank you for the correction. The typo “consitent” has been revised to “consistent” and “random” in context has been updated to  “randomly” to ensure grammatical accuracy. We thoroughly reviewed the manuscript to detect and fix spelling errors, ensuring compliance with ICML’s high quality requirements.
>
> > Q3: In line 112, “As shown in Fig. 4”, what does that mean? You may want to say “Fig. 1”.
>
> We apologize for the confusion. We want to illustrate the performance degradation of inadequate contrastive learning by “Fig.1” rather than “Fig.4”.
>
> > Q4: Subfigure needs captions as well, such as that in Fig. 2.
>
> Thank you for emphasizing the need for clearer figure captions. In the revised manuscript, we have incorporated detailed captions for each subfigure in Fig. 2.
>
> > Q5: In line 319, “further technical” should be revised to “and further technical”.
>
> Thank you for your suggestion. We have revised the phrase “further technical” to “and further technical” in the manuscript as recommended.
>
> > Q6: Why did the author use two separate models for pseudo-tag generation and padding?
>
> Some previous works that train the model using pseudo-labels generated by the same model may lead the model to accumulate error once incorrect pseudo-labels are generated [1,2]. To address this issue, we propose a dual-branch soft pseudo-label generation strategy for missing label imputation. Experimental results on Pascal07 dataset demonstrate that the dual-branch architecture exhibits superior performance and enhanced robustness compared to its single-branch counterpart. As shown in the following table:
> | Model                  | AP     | 1-HL   | 1-RL   | AUC    |
> |------------------------|--------|--------|--------|--------|
> | COME(dual-branch)      | 0.590  | 0.935  | 0.855  | 0.873  |
> | COME(single-branch)    | 0.586  | 0.935  | 0.852  | 0.872  |
>
> > Q7: In Eq. 11, whether the selection of threshold value is empirical or not?
>
> Thank you for your question. For the upper bound of the threshold $\tau_h$, we simply set it to 0.5, which is conventionally used as the threshold in classification tasks. We have conducted analysis experiments to investigate the influence of $\tau_l$ on five datasets. For pascal07 dataset, we varying the value of $\tau_l$ from [0, 0.4] with an interval of 0.1. The results are summarized in the following table:
> | Dataset \ $\tau_l$  | $\tau_l=0.4$ | $\tau_l=0.3 $ | $\tau_l=0.2$ | $\tau_l=0.1$ | $\tau_l=0.0$ |
> |-----------|---------------|---------------|---------------|---------------|---------------|
> | Pascal07  | 0.5852        | 0.5901        | 0.5904        | 0.5873        | 0.5841        |
>
> From the results, we simply set the $\tau_l$ as 0.25 for Pascal07 dataset. Intuitively, during the initial phases of training, a more stringent threshold is employed to ensure the high quality of pseudo-labels, while the threshold is gradually relaxed to extend the range of generated pseudo-labels.
>
> References:
> [1] Dual-Decoupling Learning and Metric-Adaptive Thresholding for Semi-supervised Multi-label Learning. ECCV 2024.
> [2] Debiased self-training for semi-supervised learning. NIPS 2022.
>
> Anonymous link:
> https://anonymous.4open.science/r/6513/rebuttal.pdf

---

### Official Review · Reviewer_abe3 · 2025-03-13

**Overall Recommendation:** 4

**Summary:**

This paper proposes a multi-view multi-label learning approach by integrating compact semantic information learning and pseudo-labeling imputation to address the degenerative multi-view contrastive learning and missing label issues. The authors elaborate the failure of multi-view contrastive learning in view absence and introduce the compact semantic information extraction framework. Moreover, the soft label filling approach is used to improve classification performance.

**Claims And Evidence:**

The authors claim to learn compact representations by maximizing mutual information, provide rigorous formula derivations, and demonstrate the model's superior performance through experiment results.

**Essential References Not Discussed:**

N/A

**Experimental Designs Or Analyses:**

The authors have presented solid experimental results, including tests under varying missing rates and ablation studies. However, it is essential to clarify whether identical hyperparameters were applied to all datasets. Providing comprehensive experimental details (e.g., dataset-specific hyperparameters) or releasing the source code would significantly enhance the transparency of this work.

**Methods And Evaluation Criteria:**

Experimental results demonstrate the model's robustness and superiority across diverse scenarios, with performance evaluated using standard metrics commonly adopted in multi-view multi-label classification tasks.

**Other Comments Or Suggestions:**

(Line 88) The period before “(Federici et al., 2020; Tasi et al., 2020)” should be removed. Double check for this manuscript is crucial.

**Other Strengths And Weaknesses:**

Other Strengths:
1. This paper explores an interesting problem: in practical applications, view missing and label missing are very common, and how to learn the compact semantic information between different views is critical but challenging in multi-view learning.
2. This article provides a complete explanation of motivation, theoretical derivation and experimental verification.
3. The proposed compact information learning method is simple but efficient for multi-view classification.
Other Weaknesses:
1. Some statements are not clear enough: on Equation (16) and line 625, “Since we adopt the MoE fusion strategy to model $p(z| X^V)$, we have:” which is not very clear.
2. In the paper, the pseudo label generation models are able to effectively mitigate label missingness. However, beyond the ablation studies, I recommend further investigation into the efficacy of this claim.

**Questions For Authors:**

1.I suspect that the dual-branch design could incur high computational costs (e.g., training time or memory consumption). However, the authors do not provide runtime efficiency with other models. Including such experiments would strengthen the practical relevance of this work.
2.Why do the authors introduce some methods that can not handle the missing views and labels simultaneously?

**Relation To Broader Scientific Literature:**

The paper aims to design a novel network for the incomplete multi-view missing multi-label Classification (iM3C) task, exploring the implementation of compact cross-view representation learning and dual-model pseudo-label generation. On the basis of the existing information theory and hypothesis, the paper explores the problem of contrastive learning in multi-view learning.

**Theoretical Claims:**

The authors hypothesize that classification-relevant information is embedded within the multi-view shared representations. Building on this premise, they propose to learn such shared representations by maximizing cross-view mutual information, accompanied by mathematical derivations.

---

> ### Author Rebuttal · Authors · 2025-03-30
>
> Thank you for your valuable suggestions! Below, we will address each of your questions.
> > Q1: Providing comprehensive experimental details (e.g., dataset-specific hyperparameters) or releasing the source code would significantly enhance the transparency of this work.
>
> Thank you for your suggestion. We will release the code and pre-trained checkpoints upon acceptance. For reproducibility we build on the open source implementation of COME.
> > Q2: Some statements are not clear enough: on Equation (16) and line 625, “Since we adopt the MoE fusion strategy to model $p(z\vert X^V)$, we have:” which is not very clear.
>
> We sincerely apologize for the confusing. Similar to previous work [1], we propose to factorise the joint variational posterior as a combination of unimodal posteriors, using a Mixture Of Experts (MOE), therefore we have Eq. (16).
> > Q3: In the paper, the pseudo label generation models are able to effectively mitigate label missingness. However, beyond the ablation studies, I recommend further investigation into the efficacy of this claim.
>
> Thank you for your suggestion. We conducted an in-depth investigation of the dual-branch soft pseudo-labeling strategy under varying label-missing rates on the Pascal07 dataset, with the corresponding four metrics ​presented in Figure 3 of the PDF document ​provided via the anonymous link. The experimental results demonstrate that the dual-branch soft pseudo-labeling strategy exhibits superior performance under elevated label-missing conditions, particularly when the missing rate is 70%.
> > Q4: (Line 88) The period before “(Federici et al., 2020; Tasi et al., 2020)” should be removed. Double check for this manuscript is crucial.
>
> We appreciate the reviewer’s careful attention to detail. The period before the citations in line 88 has been removed in the revised manuscript to adhere to proper punctuation conventions. Thank you for highlighting this oversight.
>
> > Q5: I suspect that the dual-branch design could incur high computational costs (e.g., training time or memory consumption). However, the authors do not provide runtime efficiency with other models. Including such experiments would strengthen the practical relevance of this work.
>
> Thank you for your constructive suggestion. We conducted a fair comparison between COME and three other deep models capable of simultaneously addressing both missing views and missing labels under identical experimental protocols, and report their training time and inference time (Unit: seconds) in the following table:
> | Dataset   | Phase     | DICNet   | SIP      | UGDPD-NET | COME (Ours)     |
> |-----------|-----------|----------|----------|-----------|----------|
> | Corel5k   | Training  | 935.65   | 166.17   | 314.34    | 877.03   |
> | Corel5k   | Inference | 0.822    | 0.896    | 0.572     | 0.873    |
> | Pascal07  | Training  | 460.41   | 301.664  | 1480.44   | 1260.50  |
> | Pascal07  | Inference | 5.096    | 5.467    | 3.572     | 4.637    |
>
> As indicated in the table, COME requires a prolonged training period on both datasets. The additional computational overhead is primarily attributed to the dual-branch architecture in our method. In future work, we plan to explore efficient alternatives to the dual-branch soft pseudo-label generation strategy to optimize computational efficiency.
> > Q6: Why do the authors introduce some methods that can not handle the missing views and labels simultaneously?
>
>  Thank you for this important question. Due to the scarcity of methods in iM3C that simultaneously address missing views and labels, we incorporated approaches handling single missing scenarios (e.g., missing views or labels) as comparative benchmarks. Specific details are provided in Appendix C. Moreover, the experimental results demonstrate that models handling individual missing scenarios exhibit suboptimal performance, further underscoring the inherent challenges of iM3C tasks.
>
> References:
> [1] Variational Mixture-of-Experts Autoencoders for Multi-Modal Deep Generative Models. NeurIPS 2019.
>
> Anonymous link:
> https://anonymous.4open.science/r/6513/rebuttal.pdf

---

### Decision · Program_Chairs · 2025-05-01

**Decision:**

Accept (poster)

**Comment:**

This paper received four effective reviews, and all of them are positive. Overall, the paper is of good quality and should be accepted.